# A Flashforward Look into Solutions for Fruit and Vegetable Production

**DOI:** 10.3390/genes13101886

**Published:** 2022-10-18

**Authors:** Léa Maupilé, Adnane Boualem, Jamila Chaïb, Abdelhafid Bendahmane

**Affiliations:** 1Institute of Plant Sciences Paris-Saclay (IPS2), Université Paris-Saclay, INRAE, CNRS, University Evry, 91405 Orsay, France; 2Vilmorin & Cie, Route d’Ennezat, 63720 Chappes, France; 3Vilmorin & Cie, Paraje La Reserva, 04725 La Mojonera, Spain

**Keywords:** outdoor farming, indoor farming, agroecology, sustainable agriculture, plant biotechnology, plant breeding

## Abstract

One of the most important challenges facing current and future generations is how climate change and continuous population growth adversely affect food security. To address this, the food system needs a complete transformation where more is produced in non-optimal and space-limited areas while reducing negative environmental impacts. Fruits and vegetables, essential for human health, are high-value-added crops, which are grown in both greenhouses and open field environments. Here, we review potential practices to reduce the impact of climate variation and ecosystem damages on fruit and vegetable crop yield, as well as highlight current bottlenecks for indoor and outdoor agrosystems. To obtain sustainability, high-tech greenhouses are increasingly important and biotechnological means are becoming instrumental in designing the crops of tomorrow. We discuss key traits that need to be studied to improve agrosystem sustainability and fruit yield.

## 1. Introduction

In 2019, the global annual temperature increased by 1.47 °C compared to the 1951–1980 normal climate period [1,2]. In 2022, the global annual temperature is forecast to continue the series of warmest years. Current agricultural practices need to be adapted to contend with the predicted extreme temperature increase [3] and the consequences of that increase on the environment [4]. This global warming is of particular concern for agriculture and food supply. Agriculture has become more invasive and destructive adversely affecting wild ecosystems, e.g., the decline of pollinator populations and pollination efficiency [5,6], in order to keep up with the worldwide increased demand for food due to global population growth.

Between 1990 and 2014, space dedicated to human occupation increased by more than 50% [7]. Agricultural areas have followed the growth tendency of the world’s population, which is estimated to reach 11 billion in 2100 [8]. Of the 149 million km^2^ of land on Earth, 104 million is habitable and half of this portion is dedicated to agriculture [2] (Figure 1).

Based on these statistics, solutions for the future of agriculture are crucial. Crop production must be designed to be more respectful of the environment while also producing more food. Meeting this goal remains a problem, especially in a sub-optimal and space-limited environment where most crops that are poorly malleable require specific growing conditions. Fruits and vegetables are good examples of such crops. Despite representing a small proportion of total crop production area compared to cereals and coarse grain (respectively 7.3% and 64% in 2014) [9], their fourfold yield (135,299 hg/ha, 189,348 hg/ha and 41,131 hg/ha, respectively, for fruits’, vegetables’ and cereals’ world production, in 2019) [2] and high added value make these crops of major concern. Fruits and vegetables are also grown worldwide using different farming methods. However, when it comes to biotechnological advances, fruits and vegetables are behind field crops.

In this review, we focused our investigation on fruit and vegetable cultivation, as they play a central role in human nutrition and health. We also considered two agricultural models: greenhouse and open field cultivations, which we referred to as indoor and outdoor production systems, respectively. We synthesized the major issues for both agrosystems and discussed the possible solutions. For the outdoor production system, we identified crop geographic distribution, biodiversity, pollination services, soil quality and water management as crucial topics. For the indoor production system, we selected the topics of inputs, energy and location, and technological and biotechnological means. From there, we emphasized the necessity of improving plant capacities for self-defense, nutrition, growth and fruit set. For all of these topics, we identified relevant keywords that we used to search and select more than 200 publications that we reviewed. Our research resulted in a comparison of cutting-edge new methodologies and existing production techniques. This allowed us to highlight promising leads for the next generation of agricultural production systems.

## 2. Solutions for Outdoor Production Systems to Reach Sustainability

### 2.1. Crop Migration and Adaptivity to Face Climate Change 

The occurrence of extreme temperatures is increasing [3]. Although this phenomenon is currently mainly observed in tropical areas, it is predicted to become a worldwide problem [10]. Soon, a temperature that is currently the average in a certain region will become the average temperature of a more northern region, as global temperature increases. King et al. predict a northward shift of climate zones of up to 1200 km from today to 2099, producing new regions of arable land in northern latitudes [11]. This newly available land could mitigate the impact of such climate variations [12,13,14] by adjusting crop geographic distribution. However, crop reallocation may be difficult, requiring major social and political changes, especially for some fruit and vegetable crops depending on geographical designation (e.g., vine production) [12,15]. Tools, such as suitability land maps, have been developed to predict the optimal geographic localization regarding these crop varieties and could be extended to fruit crop species. Implementing more environmental parameters and proposing scalable solutions would also be required [16,17]. Only a few studies on cereals and vine production have assessed the true potential of growing new crops in new regions because the value of this practice is only now being considered [11]. Crop migration is also not sufficient to counteract strong climate variations, which are occurring more frequently and can drastically compromise entire production cycles [18].

### 2.2. Promoting Biodiversity

Another factor that must be considered when transforming crop production systems is ecosystem health and biodiversity. Grassland conversion to agricultural lands, deforestation and fragmentation of natural ecosystems have destroyed the habitats of many species, leading to decreased plant and animal biodiversity [19,20]. The increase in area used for food production and the high demand for bioresources such as biofuels are leading to competition for land use between food and non-food crops [21].

Increasing the arable land area to increase food production is only a temporary solution, and research aimed at improving agricultural efficiency should be favored. The promotion of biodiversity could be conducted using growing practices such as intercropping [22] (Figure 2).

Currently, the most common intercropping systems grow cereals with legumes, as the latter improve the nitrogen level in the soil, but there are also encouraging results with assays of intercropping fruit and vegetable [23]. Agroforestry, or crops and tree co-cultivation, increases soil fertility [24], biodiversity and carbon dioxide sequestration capacity and offers biological and mechanical protection to crops [25]. However, the main barriers to agroforestry expansion are the reduction of cultivation space for crops and light occulted by the tree canopy [26]. Changing agricultural practices requires time and money for farmers and harvesting fields can be more complex in the case of intercropping.

### 2.3. Safeguarding Pollinators

Fruit yield is dependent on flower fertilization, so the decline of pollinators, which confer an annual pollination service estimated to be about USD 235–577 billion in 2016 [27], directly impacts productivity [6]. A consequence of the harmful effect of pesticides on pollinators is the significant reduction of flower visitation, which has been shown to be correlated with a lower yield of marketable fruits in many crop species [28]. It has also been reported that for some species there is flowering three days early per 1 °C increase in the monthly temperature [29]. Consequently, asynchrony is emerging between pollinators and the flower life cycle, leading to a decrease in the effectiveness of pollination [30]. In an effective agro-ecological system model, pollination should be entirely performed by an abundance of various insect pollinators, and farmers should avoid the addition of chemicals. The most well-known crop pollinators are bees, but numerous other insect families, all functionally different, are involved in plant pollination [31]. Among 100 of the most consumed insect-pollinated fruits and vegetables, bees from the Apidae, Megachilidae and Halictidae families visit the largest amount of crop species (respectively, 93, 53 and 61). Still, more than half of these crops (56) are also pollinated by Syrphidae and a third (33) by Formicidae [31]. Positive correlations between an increase in pollinator biodiversity and fruit set, seed set and fruit quality were reported in various species worldwide [32,33,34,35,36]. The total area of crops pollinated by honeybees increased by 17% between 2005 and 2010, which was more than twice the increase rate of the honeybee population during the same period, showing that agriculture is more dependent than ever on pollinators [37]. Thus, better ecological management of land use, including the promotion of floral species’ diversity and reduction of chemical use, is required to maintain the biodiversity and the abundance of pollinators [38,39,40] (Figure 2).

### 2.4. Preserving Soil Health

Healthy ecosystems also include healthy soils. Soil is the habitat of complex associations of microorganisms. These microorganisms constitute the soil microbiome and are responsible for organic matter decomposition and carbon and nitrogen cycle regulation [41]. In agriculture, soil contamination mainly comes from chemical fertilizers, which pollute soils with their heavy metal content [42] (e.g., mercury, cadmium, and arsenic). Pesticide residues also add to soil contamination [43]. Phytoremediation, or the use of plants to remove pollutants, is a solution to decontaminate soils [42,44]. However, this technic suffers from low efficiency. To reduce soil damage in agriculture, investigations on clay-humic complexes, which control the mobility of toxic metals and the nutrient availability for plants [45,46], led to the development of humic products [47,48,49] or biostimulants [50]. Technologies such as alternative fertilizers, e.g., biochar and brown coal waste, are also being tested with promising results [51,52,53]. Alternative crop practices, such as conservation agriculture, regroup other solutions to limit soil nutrient depletion and soil compaction: a reduced or an absent tillage, crop rotation [54], crop cover [55,56,57,58,59,60] and integrated nutrient management [61,62]. In fruit and vegetable production, organic farming is increasingly developed. This eco-friendly system increases soil quality [60] but remains contested regarding its impact on yield [63,64,65,66]. Because microbiome composition has a major impact on crop yields, complementing the soil microbiota could be another means to improve soil quality. Metagenomic analyses can be used to identify associations of microorganisms present in a soil sample [67]. With such knowledge, the soil could be improved by inoculating specific microorganism species [48,68] and better understanding the association between plants and beneficial microbes in the rhizosphere [69] (Figure 2). However, plant–soil interactions are complex and highly dependent on environmental parameters. External inputs can be easily diluted or leached, making soil microbiome stabilization difficult to achieve. Soil conservation and reducing chemical pollution are also major issues for improving water use efficiency and reducing freshwater pollution.

### 2.5. Preserving Water Availability and Quality

Water availability is becoming more and more of a major concern for both rainfed and irrigated growing systems. Although the intensification of drought and flooding episodes [70,71,72,73,74,75] increases crop water requirements [76,77,78] and causes water stress-associated yield loss [79,80,81], anthropic pressure is the main factor of increased water demand [82]. Crop irrigation constitutes roughly 85% of global freshwater consumption [83], and the increase in the planting area is proportional to the water demand [82]. About 70 to 96% of fruit is water [84], and, thus, its availability is a limiting factor for fruit crop cultivation [85]. Although irrigated productions avoid yield loss due to water stress conditions [86], converting all current rainfed cropland to irrigated ones would not be possible given the water availability and water requirement ratio [87]. However, the impact of water availability could be reduced if crop water use efficiency (WUEc) is improved [88]. Moreover, nitrogen fertilization and crop evapotranspiration can lead to freshwater pollution [89] and salinization [83]. A sustainable agro-ecological system should both reduce water pollution and improve its WUEc. Less-polluting crop practices should be implemented; nitrogen recycling through wastewater treatment is one possibility [89]. This could be supplemented by desalinization [90] and wastewater reuse for crop irrigation, which would a priori confer a fruit production with a similar yield and quality [91]. More studies are required, though, to assess the potential risk of chemical and biological contamination in soil and food [92]. Water savings could also be obtained by building supplementary water harvesting structures [83,93]. Improving soil infiltration and retention capacity [94] can be achieved through the management of soil conservation practices, such as soil structure-adapted tillage [95], mulching, cover crop and canopy management [96]. The limit remains currently mitigated or lacking results on the effect of such practices on WUEc. To improve plant water absorption from the soil, another promising agricultural technique is grafting—already widely extended in solanaceous and cucurbits growing—where an increase in WUEc is obtained [97,98], though conferring mitigating results in the vineyard [96].

Better evaluation of crop water needs [99] also enabled the optimization and the development of more efficient irrigation. Crop vigor has been successfully improved by regulated deficit irrigation or partial root drying [93,100] in fruit trees and vines, where even improved fruit quality and more rapid fruit maturation were obtained [96,101,102]. Still, other species such as melon endured significant yield loss with this technique [103]. Surface irrigation, the most currently used technology, could also be replaced by higher efficiency irrigation systems [83] such as drip irrigation [83,104,105] or optimized subsurface systems [106]. These systems can, however, be unadapted for crops with high irrigation needs and require regular maintenance. Precision irrigation [107] and irrigation scheduling are also arising, with the development of plant-based indicators such as sap flow or stem diameter which estimate vegetable crop water stress, and also soil-based and weather-based indicators [85,108]. These parameters can be measured with innovative sensors, but this remains challenging in a fluctuating environment [85,109]. Such technologies are thereby up-and-coming for indoor agriculture.

## 3. Solution in Indoor Production Systems to Reach Sustainability

To develop climate-independent agriculture, in the 1950s indoor production was rapidly adopted, mainly for vegetable production. The four major species cultivated in greenhouses are tomato, pepper, cucumber and lettuce [110]. Such isolated systems enable optimized plant growth, yielding more marketable fruits than those produced in fields due to the controlled environmental parameters.

### 3.1. Needs for New Sustainable Indoor Growing Systems

Greenhouse production enables the use of optimized inputs in reduced quantities. Water consumption is decreased due to highly efficient water absorption and reduced leaf transpiration in enclosed spaces [111]. It should be noted that current indoor agriculture relies on high levels of plastic use and energy consumption [112,113]. The “Sea of Plastic” in Almeria, Spain, is the most illustrative example of plastic usage; in 2009, greenhouses with plastic coverings had a surface area of about 27,000 ha in this locale [111]. Polymers offer the advantages of light diffusion and a low weight [111], and the extension of their service life could increase their sustainability. In less sun-filled regions, the problem becomes that most sophisticated structures consume large amounts of electricity to regulate lighting and temperature [112]. In addition, current indoor production locations are frequently located far from consumer areas [111,114], raising food supply prices due to transportation and contributing to CO_2_ emissions. Urban agriculture using optimized greenhouses to reconcile increased food needs and urbanization has been proposed and implemented, for instance, in Montreal with the world’s largest urban rooftop greenhouse (https://montreal.lufa.com/, accessed on 20 September 2022) [115,116,117,118]. Innovative indoor systems such as vertical farming with multiple horizontal or vertical growing surfaces could provide a high production yield and close proximity of food supply to consumers while decreasing environmental consequences [119,120]. New growing systems such as aeroponic culture, an in-air water culture with nutrient-enriched spraying [121,122], or aquaponic culture, the growing of aquatic organisms with plants mainly fed by their wastes [123,124], would then facilitate the further development of indoor agriculture. Such soilless production can achieve higher and more consistent yields with less input [121,123]. Aquaponic systems have been elaborated mainly for leafy vegetables [125,126,127], and tomatoes in some cases [128], though assessing aquatic and plant species compatibility and providing optimal crop nutrient quantity remains challenging in this system [122]. In comparison, aeroponic systems enable a strict control of crop inputs and growing conditions, a greater yield in some crops, and they have been performed on a larger diversity of plant species while requiring less expertise [121,124,129,130,131,132]. However, further research should aim for a cost reduction of this system [124]. A sustainable urban production area could combine vertical farming systems with soilless production, all using non-fossil energy sources [133] (Figure 3). Only a few crop species are currently adapted for such an isolated system, and, to date, few life cycle cost analyses of urban greenhouses have been published. Thereby further research is needed to improve productivity and establish cost-effective indoor agrosystems [134,135].

### 3.2. High-Tech Indoor Agriculture

The optimization of indoor growing depends on precise environmental monitoring. Artificially controlled environments open the possibility of growing crops in any country, regardless of the climate. High-tech greenhouses enabled the Netherlands, a country with non-optimal weather for tomato or cucumber cultivation, to reach the highest production per hectare for the two species in the world in 2019 [2]. Various types of sensors and wireless technologies enable real-time control of the general greenhouse environment, such as temperature, humidity, solar radiation, substrate pH and irrigation [136,137]. Nanotechnologies applied to agriculture are also producing encouraging results for pest control [138], fertilization [139], plant health monitoring [140] and yield increase [141,142]. Algorithms have been developed to analyze the values reported by greenhouse sensors and to assist the farmer in regulating the growing parameters [143]. Deep learning and artificial intelligence (AI) are also increasingly being exploited to diagnose plant diseases [144], monitor crop growth and control fruit harvest [145]. While the idea of producing fruits and vegetables around the clock is appealing, the cost of production and environmental implications must be thoroughly examined. Some emerging technologies require hazardous base materials to manufacture and use. They can be also expensive to produce and energy intensive. Recycling them also adds new issues. Photovoltaic panels, for example, have a 25–30-year average lifespan and will be responsible for about 78 million tons of waste by 2050. Recycling the highest proportion of photovoltaic components, which are aluminum, silver, glass, copper, and rare metals (e.g., indium and gallium), although costly, is possible. However, from the recycling residues remain heavy metals (e.g., cadmium and lead) coming from batteries and solar cells, which then become hazardous waste, thereby being of major concern [146,147,148]. Another consideration is that little is known about the potential risks posed to plants [149], ecosystems or human health [138] when nanomaterials are released into the environment. According to current knowledge, nanoparticles can cause phytotoxicity due to crop uptake, translocation and accumulation into plant tissues and biotransformation of these highly reactive particles [139,150]. They can also decrease soil biomass [151,152,153], and the first safety margins are only being evaluated in some products [154]. As a result, comparative studies of the life cycle of these technologies and their impact on the environment are required to assess their utility.

### 3.3. New Crop Traits for Indoor Farming

Taking all of this into consideration, identifying plant needs for indoor growth seems to be a key point in enhancing productivity. Indeed, growing in restricted and isolated spaces requires new characteristics of fruits and vegetables. First, to grow plants in a smaller area, limiting crop size without lowering yield is necessary [120]. Second, rapid growth is desired to reduce the time to reach fruit maturity [120,155,156]. Third, in a controlled environment, plants should face fewer biotic and abiotic stresses but should still have optimized disease and pest resistance to avoid yield loss. The last standard for indoor crops relates to consumers. Growing fruits and vegetables with a limited shelf-life in an urban growing environment would shorten the commute from the production zone to the market, reducing waste [155]. Consumers also expect homogeneous product appearance (e.g., size, color, shape, firmness and absence of external defects) and taste [157,158,159], which would be more easily obtained inside a controlled environment [160,161].

## 4. Genetic Improvement of Cultivated Fruits and Vegetables

For decades, breeding and genetic engineering have been providing multiple possibilities for plant improvement, and current investigations could make vegetables and fruits more suitable for growth in future agriculture systems.

### 4.1. Improving Plant Defense Mechanisms

Plant defense mechanisms are the most extensively studied subject of plant biology (Figure 4a). The number of patents relating to improving plant resistance outnumbers those relating to other biological features (Figure 4b). Almost all transgenic plants cultivated today are modified for disease or herbicide resistance, owing to the economic costs associated with crop losses caused by pests and pathogens. These losses can be as high as 30% of the yield of some of the world’s most important crops [162]. Pesticide use, on the other hand, almost doubled between 1990 and 2018 [2]. Genetic resistance could be a leading solution for reducing chemicals in agriculture. Resistant plants could be developed by conventional breeding or biotechnological means. The most famous type of engineered resistance is Bt crops, producing the *Bacillus thuringiensis* toxin that is lethal to insect pests. Genome editing has also led to the development of effective resistance against powdery mildew [163] in tomatoes, potyviruses in cucumbers, and citrus canker [164]. Despite many breakthroughs, genetic resistance is often broken by emerging strains, highlighting the need for polygenic resistance.

Crop tolerance against abiotic stressors, such as extreme temperatures, flooding, drought and soil salinity [165], also needs to be developed. Thermo-tolerance has been obtained by regulating proteins (e.g., heat shock proteins) or using microRNA [166,167,168]. Transgenic plants such as tomatoes [169] and cabbage [170] with improved water absorption [170], leaf transpiration regulation or antioxidant enzyme levels [169] have demonstrated enhanced drought tolerance abilities. Similarly, salt stress tolerant vegetables [170,171] and fruits [172] have also been developed. The ability of a plant to cope with environmental variation, known as plasticity, is a trait that has not been well investigated in crop plants. Root plasticity has been improved in some species [173], but plasticity that extends to all plant organs still needs to be investigated. Given that climate change is expected to have an impact on plant disease resistance [174], it is instrumental to develop plants that are resilient both to biotic and abiotic stresses.

### 4.2. Improving Plant Nutrition Mechanisms

Improving the root system is important to enhance plant performance since root absorption capacity is vital for increasing nutrient and water intake (Figure 4c). Although increases in root hair biomass have been studied primarily in cereals, a correlation between increased root density and a higher yield was reported in some fruits and vegetables [175,176] so this trait needs further study. Stomates, which regulate water transpiration and gaseous exchange, also significantly impact water use efficiency and resistance to pathogenic bacteria [170,177,178]. Although it has been found that the acceleration of stomata opening and closing is correlated with increased water use efficiency and growth [177], research is mainly concerned with reducing stomate numbers or aperture to reduce transpiration and fight abiotic stresses [170,178]. Another interesting area in the field of agricultural engineering is the improvement in a crop’s photosynthetic capability. Photosynthesis is a chain of chemical reactions catalyzed by various enzymes and initiated by photosystem excitation. Improving the carbon dioxide (CO_2_) uptake by plants is currently being studied as a means of increasing energy conversion. It has been found that increasing the concentration of the most influential enzymes involved in controlling the CO_2_ flux, combined with a reduction in photorespiration, allows for higher CO_2_ assimilation [179]. In the model crop *Nicotiana tabacum*, an increase in the plant’s photosynthetic capacity resulted in a biomass increase of up to 37% [180]. Encouraging results were also reported in tomatoes [181] and lettuce [182]. Last but not least, plants with improved photosynthetic capabilities would also contribute to more carbon dioxide sequestration. Recent studies bring up the potential of enhancing crop–root relations to reduce greenhouse gas levels [183,184,185].

### 4.3. Growth Optimization for Space-Limited Environments

Semi-dwarf cereals have been developed for many years for lodging resistance [186], contributing to the Green Revolution. Fruit and vegetable plants could be dwarfed to better suit indoor farming (Figure 4c). Recently, Kwon et al. made a promising CRISPR-Cas9 gene edit in the tomato *SELF PRUNING* gene, responsible for short internode tomato plants. This mutant has a compact and short stature, rapid growth cycle and early flowering, making it suitable for indoor vertical farming [156]. A short stature and short life cycle characteristics were also obtained in a vine mutant [187], and comparable abilities have been investigated in melons [188], peanuts [189] and even in fruit tree species such as kiwi [190]. Currently, mainly leafy vegetables are grown indoors. More research is needed to see if new crops can be easily adapted. The elimination of possible secondary effects on the compacted plant must also be verified, such as reduced photosynthetic capacity [191] and increased pathogen development.

### 4.4. Improving Flowering and Fruit Set

Plant reproduction efficiency is another key parameter in improving crop productivity. A long flowering duration, which ideally should be independent of environmental conditions, the optimal quantity of flowers, and a high level of fertile flowers and fruit set are among the traits that have been investigated to improve fruit yields [192,193] (Figure 4c). For monoecious plants that develop separate female and male flowers, a positive correlation has been found between the number of staminate flowers and the fruit set yield. Consequently, controlling crop sex ratio is another major goal for optimizing fruit production in several species [194]. Another example of a flower-related trait impacting crop yield is flower fertility. Various factors can alter fertilization and lower the fruit set yield. Low pollen fertility or pollen germination [195] under adverse temperatures and incompatibility between pollen and pistil [196] are examples that reduce fruit set. Improving flower fertilization and fruit set under a wider range of temperatures can help achieve year-round cropping. However, flowering remains a complex mechanism that depends on numerous factors, such as plant nutrition, photoperiod and temperature. All of these factors must be considered when designing a plant with an increased fruit set.

### 4.5. Parthenocarpic Crops as a Solution for Both Outdoor and Indoor Growing

Parthenocarpy is the ability of a plant to produce fruits without ovule fertilization. Since pollinators and pollen are vulnerable to adverse weather conditions, both indoor and outdoor farming would benefit from parthenocarpic plants [197,198,199,200]. In particular, parthenocarpic cultivars could be a great choice for indoor farming systems because greenhouses are often insect-proof. Parthenocarpic fruit development could be induced by the exogenous application of hormonal treatments (e.g., N-(2-chloro-4-pyridyl)-N’-phenylurea (CPPU) [201,202] and gibberellic acids (GA) [201,203]). However, such practices may be unwelcome in an era when reducing the use of chemicals in agriculture is a worldwide objective. Breeding and genomic knowledge have enabled the selection of parthenocarpic fruits and vegetables without exogenous treatments, and some of these products are currently available on the food market (Table 1). Species such as bananas naturally exhibit obligatory parthenocarpy, and varieties such as Cavendish [204] were originally selected from parthenocarpic plants only capable of vegetative propagation. Other crops exhibit facultative parthenocarpy, which allows both sexual reproduction and parthenocarpic fruit set, in absence of ovule fertilization. In outdoor agriculture, facultative parthenocarpy would ensure a stable yield in poor climate conditions and remain compatible with insect pollination. Induced mutations leading to parthenocarpy have been identified (e.g., mutation of pat genes in tomatoes) [205]. Transgenic cultivars have also been created to optimize parthenocarpic performances (Table 1). For instance, *DefH9-iaaM* gene fusion, which induces parthenocarpy via auxin promotion, has been used in strawberry, raspberry [206], eggplant [207] and tomato [208] plants. Positive characteristics have also been reported in parthenocarpic crops. A yield increase was correlated with parthenocarpic abilities in several species, both indoor and outdoor [206,207,209]. Firmer fruits [210] and earlier fruit production [211] were also reported to be qualities induced by parthenocarpy. Additionally, parthenocarpic tomatoes have even been shown to enhance fruit set rates under extreme temperatures. However, issues such as fruit malformation [212,213] or reduction of fruit size and weight [214,215] have been also reported and should be addressed in future works. The use of parthenocarpic crops, which do not require chemicals, would greatly benefit both outdoor and indoor agriculture. The development of more commercial parthenocarpic varieties in the future is expected.

## 5. Conclusions

As described above, a combination of the various technical and genomic methods currently available is necessary in order to seriously develop sustainable agriculture. Developing indoor production systems that do not encroach on habitable lands and editing leader alleles enhancing yield and resilience will likely contribute to the required sustainability. Specifically, the ideal fruit crop for future indoor agricultural endeavors should include characteristics such as a rapid life cycle to improve productivity per year, a short stature to fit in space-limited growing areas, an efficient nutrition system to lower chemical inputs, and an optimized flowering and fruit set to ensure a high fruit yield. As an example, some parthenocarpic cucumber varieties already have a high yield of marketable fruits and low side-shoot growing, which reduces plant width and the need for pruning (e.g., Corinto [213]). Cucumbers also have a naturally short life cycle of about 50 to 90 days [226]. In this regard, the cucumber plant can serve as a model crop for fruit plant adaptation to indoor breeding. For field crops, the challenge of sustainability is more complex. Outdoor farming needs to reduce chemical inputs while maintaining high productivity. Establishing soil management means to prevent agricultural land from becoming poorly productive and improving crop resilience to cope with the deterioration of growing conditions that will only be exacerbated by global warming are other major challenges. Outdoor sustainable agriculture cannot exist without investigating crops’ ecological functions. Although much is known regarding the impact of agricultural systems on ecosystems, little is known about how crops’ ecological functions can be identified and applied to develop sustainable agriculture. In this regard, we have an open field of research that needs to be investigated. Since our review considered agricultural systems globally, more specific work is also required to identify regionalized issues and solutions for each agricultural system. Competition for land use between livestock, bioresources and crop production should also be studied to consider the needs of all agricultural systems.

## Figures and Tables

**Figure 1 genes-13-01886-f001:**
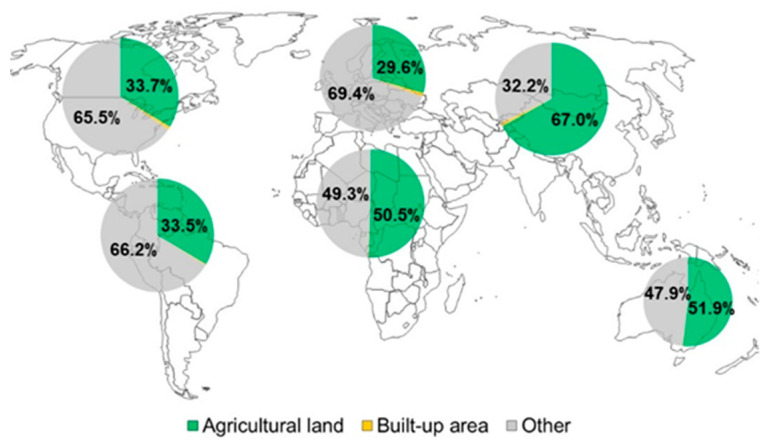
The increasing competition for land use. Habitable land use in 2018. The surface data come from FAO (www.fao.org/faostat/en/#data, accessed on 27 November 2021). The values were calculated for North America, South and Central America, Europe, Africa, Asia and Oceania, relative to the total habitable land area, which was the sum of agricultural lands, built-up areas, and non-aquatic forests and shrub areas that are not regularly flooded. The “Other” section contains forests and shrubs that are non-aquatic or regularly flooded. The “Built-up area” section represents 1% or less of the habitable land use for each geographic region.

**Figure 2 genes-13-01886-f002:**
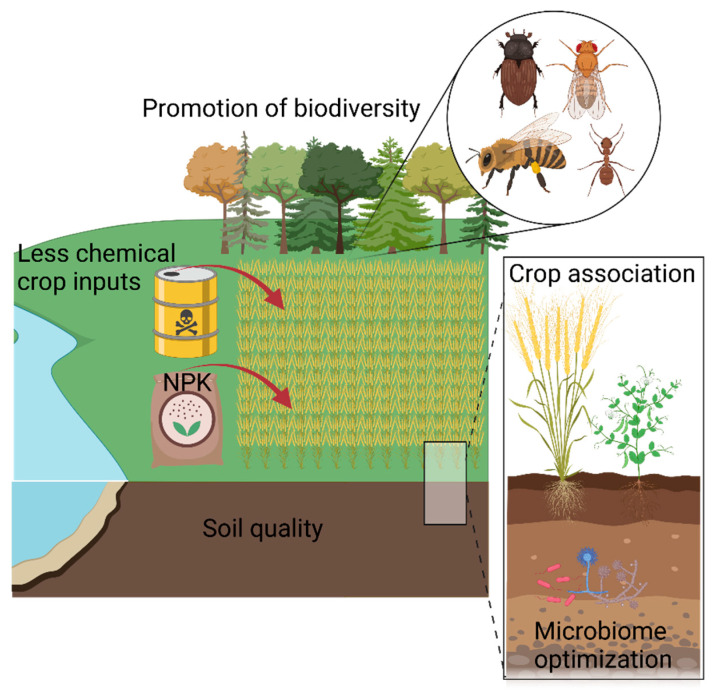
A sustainable agro-ecological system. The promotion of biodiversity includes the protection of various pollinator species and plant species in a given agro-ecological system. The reduction in chemical inputs implies that pest control and the use of fertilizers should be replaced by genetic improvement and alternative crop practices. Created with (BioRender.com).

**Figure 3 genes-13-01886-f003:**
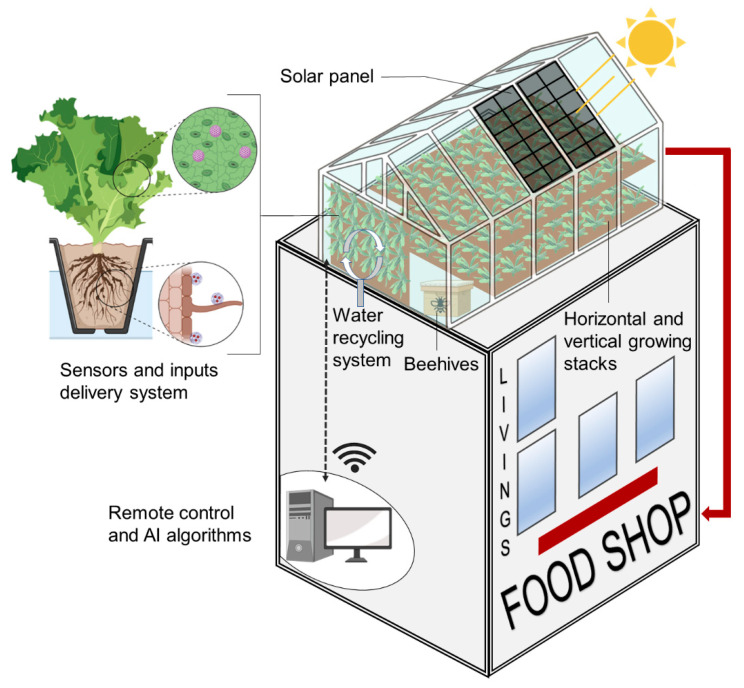
Model of an urban agriculture system. The scheme represents an indoor high-tech growing structure located on top of a building containing accommodation and a food shop. It highlights the benefits of indoor agriculture in an urban area, which includes avoiding food exportation. The sensors and input delivery system illustrate the use of nanofertilizers (on the roots, in gray), nanosensors and nanopesticides (on the leaves, in purple). The curved arrows symbolize the reduction in water transpiration losses. The solar panel illustrates the use of non-fossil energies; the beehives, an increase in insect pollination; and the red arrow, the short transition from the production area to consumption places. Created with BioRender.com.

**Figure 4 genes-13-01886-f004:**
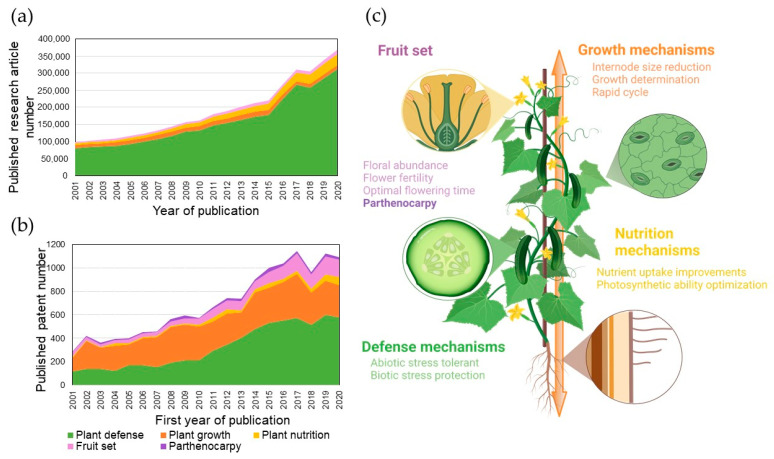
Genetic improvements to crops needed for sustainable agriculture. (**a**) The number of published patents and (**b**) research articles related to genetic improvement of fruits and vegetables and (**c**) traits to target in plant breeding. Created with BioRender.com. The data come from the Orbit Intelligence patent database (https://intelligence.orbit.com, accessed on 29 April 2021) and Web of Science (https://www.webofscience.com, accessed on 29 April 2021).

**Table 1 genes-13-01886-t001:** List of plant species with parthenocarpic fruit development currently on the market.

Species	Germplasm Name or Mutation Description	Source of Parthenocarpy
Apple	Spencer Seedless, Wellington Bloomless, Rae Ime [216] Wickson [217]	Natural
Banana	All consumed bananas are parthenocarpic. Examples of variety names: Blue Java [218] and Cavendish [204]	Natural (obligatory)
Cucumber	Camaro, Kalunga, Katrina, Socrates, Manny, Manar, Jawell, Picolino, Taurus, Tasty Jade, Tasty Green, Corinto, Lisboa, Alcazar, Sweet Success [213]	Natural
Eggplant	Talina, Galine [219]	Natural
Satsuma mandarin	Seedless Fallglo [220]	Induced mutation
Clementine	Marisol, Clemenules [221]	Natural
Grape	Corinto bianco [222], Black Corinth [223]	Natural
Grapefruit	Henderson, Marsh, Redblush [219]	Induced mutations
Pepper	Shishitoh [224]	Natural
Summer squash	Whitaker [225]	Natural
Sweet orange	US Seedless Pineapple [220]	Induced mutation

## Data Availability

Not applicable.

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
