# Peer review of "A Flashforward Look into Solutions for Fruit and Vegetable Production"

_genes, 2022, doi:10.3390/genes13101886_

Round 1

Reviewer 1 Report

Dear Authors,

Congratularions for your manuscript.

It is quite interesting.

However, I imagine that some improvement must be done before publication.

Please, see my comments and suggestions.

I hope they would be useful.

Best.

Review – Genes 1919572

Manuscript: A flashforward look into solutions for sustainable agriculture

General Comments:

The manuscript provides an interesting discussion about potential practices to reduce the impact of climate change in crop yield. Also, the authors discuss some challenges for indoor and outdoor agricultural systems to improve their yield performance.

The manuscript is well organized, but as a “review paper”, I think that it is quite short and superficial. In lot of passages, the authors only states general information or perceptions, but provider neither evidence nor references to support their arguments.

Moreover, the manuscript should consider regional contexts for the general analytic division proposed. For instance: the indoor production systems challenges, possibilities, opportunities and limitations discussed are applicable for urban areas in developed and underdeveloped countries? Similar issue may be considered for outdoor production systems. I imagine that the main question of the manuscript may be require regionalized answered.

Also, even though the authors indicated that the review is focusing on fruit and vegetable cultivation, I imagine that they must consider the interrelationship between agriculture and livestock regarding land use. This issue is critical, particularly for outdoor production systems.

Specific comments:

Lines 84- 88: The authors should expand this discussion; for instance, highlighting the huge potential of agroforest systems to improve biodiversity while increase production and preserve environmental resources.

Lines 99 - 101: This issue is very relevant. Considering that this is a “review paper”, the authors must provide more evidences highlighting the crucial role of pollinators, particularly bees, to agriculture performance.

Lines 118 – 119: The sustainable intensification literature provides lot of examples about this issue. The authors must provide more references and examples of strategies to limit soil nutrient depletion or recover degraded areas. The authors should include in the item 2 (outdoor production systems) a subtopic about agriculture and water. This resource is vital to agriculture and has been suffering high anthropic pressure worldwide. Moreover, agriculture is responsible for water pollution in many regions around the globe.

Lines 146- 147: The authors must provide examples of this situation to support their statement.

Lines 150 – 153: This is an interesting issue and a disruptive technology. The authors must provide more information about it.

Item 3.1 and 3.2 present the same writing

Lines 182 – 185: These issues are very important. The authors must provide more references and examples.

Lines 198 – 199: The authors must provide more references and examples to support their statement.

Lines 249 – 258: The photosynthetic capability is crucial not only for energy conversion and increasing yield, but for carbon dioxide sequestration. The authors should explore this perspective and provide more reference about this issue.

Reviewer 2 Report

This manuscript claimed to “review potential practices to reduce the impact of climate variation and ecosystem damages on crop yield, as well as highlight current bottlenecks for indoor and outdoor agrosystems” (Line 12-13), with a focus on “fruit and vegetable cultivation” (Line 50). Row crop sustainability has been extensively studied and reviewed whereas fruit & vegetable systems are underrepresented. This manuscript could be of value. However, in the main body of the manuscript, the authors presented brief sections (5-20 lines of texts per section) on both challenges in crop --- mainly row crops with a bit touch on vegetable and fruits --- production and possible solutions for the challenges. 

My first concern of this manuscript is about the language and style (length & organization). 

·      The authors intend to focus their  investigation on fruit and vegetable cultivation, as they play a central role in human nutrition and health (Line 50-51). But this important information is not in the Title or the Abstract. 

·      Though the main selling point of this manuscript is fruit & vegetable sustainability, a majority of the manuscript is about row crop production. It is not clear to me what is special about fruit & vegetable production system. 

·      Indoor verus outdoor system is not intuitive to me. What about using greenhouse for indoor, and using open field for outdoor?

·      As a review paper, 5-20 lines per section is short.

·      There needs to be a methodology section after the Introduction. How was the literature review done? More on this later.

My second concern is about the content. The authors mentioned they reviewed potential practices for vegetable and fruits sustainability, but the section titles and the content under each section do not convey such information. For example, Section 2 is titled as “Outdoor production system”. This title tells me nothing about the content under the section. Similarly, section 2.1 title “reallocating crops” seems to be a problem than potential practices to reduce impact of extreme temperature. The texts under this section discussed (1) warming temperature and climate change, and (2) potential crop migration and the infeasibility of migrating crops to areas that are now too cold to produce. What technology and practice is addressing what issue here? How would the technology/practice address fruits and vegetable sustainability? Similarly, almost all the sections described confusing challenges that are more related to row crops and provided very little useful information about the how technologies that could specifically address the fruit & vegetable sustainability.

Last but not the least, a robust review paper needs a robust, and most likely systematic, review methodology. Two of the papers below help explain different types of review and the methodology associate with it.

Tricco, A. C., Lillie, E., Zarin, W., O’Brien, K., Colquhoun, H., Kastner, M., Levac, D., Ng, C., Sharpe, J. P., and Wilson, K. (2016). A scoping review on the conduct and reporting of scoping reviews. BMC medical research methodology 16, 1-10.

Munn, Z., Peters, M. D. J., Stern, C., Tufanaru, C., McArthur, A., and Aromataris, E. (2018). Systematic review or scoping review? Guidance for authors when choosing between a systematic or scoping review approach. BMC medical research methodology 18, 1-7.

This manuscript lacks of robust methodology and concrete evidences or discussions on practices that would address fruits and vegetable cropping system. 

Round 2

Reviewer 1 Report

Dear Authors,

Thank you for consider my suggestions.

The new version present considerable improvements.

Best

Reviewer 2 Report

I was pleased to see the improvement in the methodology and much more comprehensive review.